# Urban Overheating Mitigation Strategies Opportunities: A Case Study of a Square in Rome (Italy)

Gabriele Battista [1,*], Emanuele de Lieto Vollaro [2], Luca Evangelisti [1] and Roberto de Lieto Vollaro [1]

1   Department of Industrial, Electronic and Mechanical Engineering, Roma TRE University, Via Vito Volterra 62, 00146 Rome, Italy
2   Department of Architecture, Roma TRE University, Via della Madonna dei Monti 40, 00184 Rome, Italy
*   Correspondence: gabriele.battista@uniroma3.it

**Abstract:** It is well-known that the occurrence of urban heat islands (UHI) is related to climate change and urbanization. Urban locations experience local overheating throughout the summer, which is uncomfortable and it has a detrimental impact on buildings ability to consume energy. In this study, a methodology was developed to assess the urban heat island effect in a localized urban area and to evaluate the effects of different kind of mitigation strategies. The numerical model was developed using the ENVI_met tool and it was calibrated with weather data and albedo measured inside the examined area and near the simulated domain. The procedure adopted overtaking the limit of the software in reproducing accurate weather conditions without calibration. Finally, combination of extensive mitigation strategies (cool pavements, greenery, grass pavers) with local strategies (shading) were investigated. An experimental and numerical investigation of a square in Rome was exanimated to evaluate the possible solution for mitigating outdoor air temperatures. Results of the paper affirm that an intervention on the pavement albedo and the increase of vegetation inside the square, lead to an improvement of the air thermal conditions. In particular, the application of the grass pavers would lead to the greatest benefits. A maximum decrease in the air temperature of 1.2 °C was obtained through the application of cool material, while the reduction reaches up to 2.88 °C when a grass paver is applied. The improve of the vegetation can bring to a maximum decrease of 1.46 °C, while the use of shading projecting roof allows a reduction up to a maximum of 2.07 °C.

**Keywords:** overheating; urban heat island; mitigation strategies; ENVI_met; model calibration

## 1. Introduction

Climate changes and urbanization are the main causes of the urban heat island phenomenon. Urbanization can be related both to the raise of the urban population despite of the countryside one, and to, deforestation and soil consumption. The urbanization rate can be computed making a proportion between the urban and the rural population. Urban growth doesn't lead to land expansion, but the increase of the people that live there [1].

People moved from the countryside to the cities all over the world [2]. In the last few centuries, the growth has started going faster, moving from 2–3% at the beginning of the XIX century to 30% in the middle of XIX century, until 2017 when urban population overcame the rural one [3].

### 1.1. Urban Heat Island Effects

Urban growth led to the urban heat island phenomenon [4,5]. Urban Heat Island represents a microclimate phenomenon of cities [6]. It was defined for the first time by Luke Howard in "The Climate of London" study of 1818 [7] as the raise of temperatures in cities compared to what happened in the rural areas. This phenomenon is higher when wind flows are reduced. The temperature difference between cities and rural areas can range between 0.5–3 °C and it can be caused by many factors, such as the high solar

radiation absorbance on concrete and asphalt that release the heat of the day during the night [8]; reduced green areas, natural and permeable surfaces; thermal human sources related to transport and air conditioning systems; low soil water retention and low level of evaporation.

Urban heat island can impact on people's lives causing many side effects [9,10]. Air and water quality are strictly related to this condition. Moreover, it leads to a more frequent use of air conditioning systems due to the increase of the cooling in buildings [11,12], causing in turn a higher use of electricity [13,14] and leading to high levels of pollutant emissions in the urban context [15,16].

### 1.2. Urban Heat Island and Landscape Relationship

Over the past ten years, UHI has rapidly risen as a consequence of changes in landscape, increasing urbanization, and a reduction in plants and water resources [17–19], which result in higher land surface temperature (LST). In particularly, factors influencing the UHI phenomena include the lack of greenery in urban areas, the characteristics of urban materials, the geometry of cities, anthropogenic heat, the environment, and location. The value of remote sensing satellite data for analysing the links between urban landscape pattern and LST has been demonstrated in several UHI research [20,21]. According to Zhang et al. [22], urban expansion in Shanghai have altered the surface energy balance and raised the sensible heat flow due to a reduction in plants and water bodies resources, reducing the difference between the LST of the city and the urban periphery. According to Mallick et al. [23], Delhi's commercial and industrial zones saw LST values that were 4 °C higher than those in the suburbs. Human activities and changes in plant density are all responsible for this.

It is acknowledged that the UHI effect is significantly impacted by impermeable surfaces replacing natural land [24]. Buildings are an essential part of the impervious surface and a significant factor in the UHI [25]. A popular area of research in recent years has been examining the connections between the buildings' three-dimensional (3D) landscape measurements and the UHI. According to Hu et al. [26], the height and design of Beijing's buildings will have an impact on surface albedo, which in turn will have an impact on UHI.

### 1.3. Urban Heat Island Mitigation Strategies

Studies on UHI mitigation strategies, such as those using green roofs [27,28], cool materials [29], vegetation [30,31], and water sources [32,33], have received a lot of attention recently. Adopting mitigation techniques is crucial for both large urban regions like cities as a whole and smaller localized urban locations like urban canyons [34,35].

To mitigate the UHI impact by lowering the pavement's surface temperature, reflecting pavements are one of the most researched and cost-effective mitigation strategies [36,37]. In principle, two factors can be altered to make a pavement more reflective: the pavement's colour and the roughness of its surface [38]. Most research indicate that making pavement surfaces whiter or lighter in colour is the most effective and practicable way to reduce the UHI impact [38,39]. Because of its innate potential to dramatically lower surface temperature because of the higher albedo and enhanced degree of thermal emissivity, reflective pavements can also be referred to as cool pavements [38].

Larger research revealed a radical shift in urban heat island reduction due to the growing usage of cool and green roof technology. According to Georgakis et al. [40], lowering wall surfaces by 2–3 °C and raising the albedo value of pavements at ground level can result in a 7–8 °C decrease in surface temperature. The study discovered a 1 °C drop in the urban street canyon's total air temperatures. When cool materials and green permeable surfaces were used in place of the old pavements and roofs in a part of Mestre, Venice, Peron et al. [41] discovered that the temperature dropped by about 4 °C. UHI effects are also influenced by pavement [42,43].

Due to their inherent capacity to cool a pavement through water evaporation, evaporative and water retentive pavements have received a great deal of research attention in addition to reflecting pavements. The reduction of stormwater runoff is essential because rainfall dictates how well a permeable, porous, and water-retentive pavement functions. Pavements that evaporate might be categorised as porous, permeable, or pervious.

Porous pavers are used to create pavement of roads. These pavers are made of a grid of cells packed with materials that may trap moisture in the perforations. The best infill for these pavers is grass since it has a greater albedo than other fillers (like soil) and because its transpiration promotes evaporative cooling [44].

Permeable pavement has a permeable top layer that directs rainwater flow into channels so that water may move around the pavement while yet allowing it to pass through straight to the base [44]. According to research, permeable pavements may effectively reduce urban problems like the heat island effect and can help reduce stormwater runoff [45].

A pervious paver is a specific kind of concrete that has a high porosity level and permits water to permeate the pavement directly. According to studies, maintaining water near the pavement's surface is essential for enhancing the pavement's thermal properties. This can be done by increasing a pavement's capillary activity or by misting water over the pavement surface [46].

Reverting to the previous environment is one of the simplest ways to reduce the impact of UHI. This is since the earlier-mentioned decline in vegetation and the increased predominance of hard materials and structures are major contributors to the UHI impact [47–50]. In addition to providing shade for people and buildings, more trees also slow down the wind under canopies and cool the air through evapotranspiration [48,51]. Urban green spaces have cooler air and surface temperatures than the surrounding metropolis, therefore even a modest amount of green space can have a cooling impact [52,53]. The neighbouring cityscape benefits from this cooling impact as well.

### 1.4. Urban Heat Island Studies in the City of Rome

The impacts of UHI on Rome's climatic conditions and the influence of mitigation measures, specifically in terms of lowering air temperature values in specific metropolitan areas, are the subject of several research and findings that have been published in the literature [54–56]. Cecilia et al. [57] studied the summer UHI in Rome from 17 weather stations network from 2019 to 2020, while Battista et al. [58] studied the phenomenon along the whole year with 23 weather station inside the city finding correlations of UHI phenomenon and the land characteristics. According to an analysis of three year's worth of data from 2015 to 2017, Rome's UHI is more severe in the summer than it is in the winter, with values of 0.7 °C and 1.0 °C, respectively [56]. To determine the effects of various UHI mitigation measures on air temperatures and outdoor thermal comfort conditions, Battista et al. [59] investigated a heavily crowded Roman square. In order to assess outdoor areas that are essential for reducing the impacts of UHI, Salata et al. [60] conducted a field survey inquiry in Rome. Another study evaluated various UHI mitigation strategies in Rome using monitoring data and numerical analysis, and the results showed that new urban development increased the average ambient temperature by up to 3.5 °C during the midday, while using reflective materials and vegetation could lower the temperature by up to 2 °C [61]. In their study of the effects of heat waves on three Rome neighbourhoods from 2015 to 2017, Zinzi et al. [55] discovered that while cooling energy consumption in buildings increased by 87% during these times, the UHI index might rise by up to 1.5 °C as a result. Morini et al. [62] finds that in the city of Rome that an albedo increase leads to the decrease of the 2-m air temperature at day-time and at night-time by up to 4 °C [61]. A numerical model of a university campus in Rome was developed by Salata et al. [63] and it was used to examine various UHI mitigation strategies to reduce the outdoor air temperature of the site. Ilaria and Susca [64] analyse extensive green roof, living wall and green façade deployment finding that the UHI mitigation increases by 70% and 90% augmenting LAI from 1.5 to 3 for green roofs and living walls, respectively.

Highly reflecting materials [65], green roofs [66] and urban greenery [67] are common tactics in localized urban areas in Rome. The UHI index may be influenced by considering the urban features of a historic city like Rome, such as its winding alleys that form urban canyons and its traditional roof materials made of low solar reflectivity Roman clay tiles [68,69].

### 1.5. Aim of this Study

The objective of this work is evaluating several technical solutions for reducing the microclimatic conditions in a heavily populated Roman neighbourhood, the Mancini Square in Rome (Italy). In order to evaluate the impact of the mitigation strategies in terms of a drop in air temperature, a numerical model was built through the ENVI_met tool [70]. To establish the boundary conditions and calibrate the numerical model, air temperature and relative humidity data were collected.

The manuscript is structured in this way: Section 2 provides information about the methodological approach, the case of study and the suggested mitigation techniques, also considering the experimental measurement campaign and the generation of the numerical model; the obtained results are shown and discussed in Section 3; finally, Section 4 draws the conclusions.

## 2. Materials and Methods

### 2.1. Methodology

The core of this study is the analysis of the selected area in order to verify the effectiveness of different countermeasures aimed at reducing the urban heat island effects. The applied methodology is based on the following steps and shown in Figure 1:

1.　Selection of a densely populated urban area characterized by few green areas;
2.　Assessment of the Urban Heat Island Intensity (UHII) able to justify the choice of the urban area chosen in the point 1. The UHII is calculated as the maximum daily difference of the outdoor air temperature of the examined area and a reference one that must be a rural area near the city. The values are calculated as the following:

$$UHII = daily\_maximum\ (T_{UA} - T_{RA}) \tag{1}$$

3.　Identification of the more effective strategies, starting from the best practices applied in other case studies;
4.　Experimental monitoring, taking into account the geometry, the albedo of the buildings and environmental conditions in terms of temperature, rainfalls, relative humidity, solar radiation and wind speed;
5.　Numerical modelling generation and calibration using ENVI_met;
6.　Mitigation techniques analysis through the use of numerical simulation. The data related to the improved scenarios will be compared to the ones of the initial configuration.

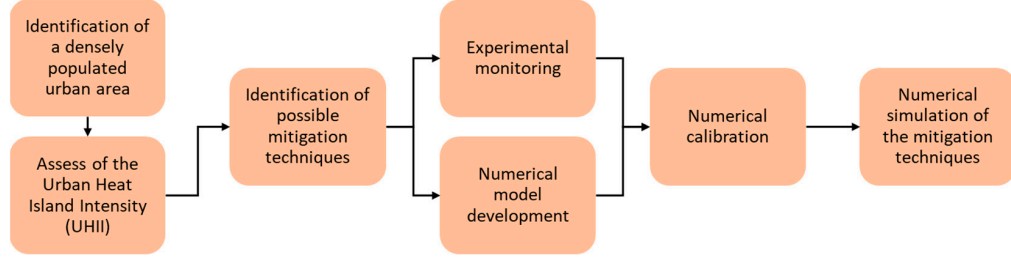

**Figure 1.** Flowchart of the methodology.

The numerical model was developed by the use of ENVI_met tool and it was based on the following steps:

1. Implementation on the geometrical characteristics of buildings, trees, vegetation, roads and soils throughout the use of Google Earth aerial view of the zone;
2. Setup the material characteristics of the building envelope, road, soils, trees and vegetation, with also the implementation of the on-site albedo measurement;
3. Setup the initial conditions of data used for the numerical simulations, and setup the forced input of air temperature and relative humidity conditions of the boundary with data acquired by a weather monitoring station placed near the simulation domain;
4. Numerical model calibration step that consists of checking, simulation after simulation, the relative error of simulated and measured data where the weather station was installed. In each simulation, to minimize this error, it is possible to correct point by point the forced input data of the numerical model related to the air temperature and relative humidity. The difference between the experimental data and the simulated ones were assessed using two statistical indices. The quantitative indicators [61] used in this work are, taking into account the model-predicted $P_j$ and observed variables $O_j$ at each instant $j$, and the amount of examined data $N_D$, respectively:

   a. Mean Bias Error (MBE): it shows if the model values exceed or undervalue the observed data. Because they set off against one another, this indicator does not function properly when anticipated values are alternately overestimated and underestimated.

$$\text{MBE} = \frac{\sum\limits_{j=1}^{N_D} (P_j - O_j)}{N_D} \qquad (2)$$

   b. Mean Absolute Error (MAE): it is similar to the MBE indicator, but accounting for the absolute difference between expected and actual values. Therefore, this indication is helpful when the projected values alternate between being overestimated and underestimated.

$$\text{MAE} = \frac{\sum\limits_{j=1}^{N_D} |P_j - O_j|}{N_D} \qquad (3)$$

5. Implementation of the different mitigation strategies to be used and assessment simulation in order to evaluate the variation of the thermal conditions inside the objected area of the study.

### 2.2. The Case Study

Mancini Square in Rome serves as the case subject for this work. Wide asphalt zones that are employed as sidewalks and pavement make up the research area. Additionally, there are low trees that do not provide enough shade inside Mancini Square. Due to the dense population in this area, the effectiveness of the mitigating measures is maximized. Figure 2 depicts an aerial view of the modelled area in the Flaminio district, along with a location indication for the weather station where the air temperature and relative humidity were measured and the Mancini and Carracci squares. This neighbourhood is adjacent to both the Tiber River and the heart of Rome. Mancini Square has a surface area of about 6000 m$^2$ and it is located at latitude 41°55′51.1″ N and longitude 12°27′51.9″ E. The examined area is located between the city centre and the North city end. In particular, the distance of the examined area and the city centre is 4.6 km, while the end of the city is 6 km far.

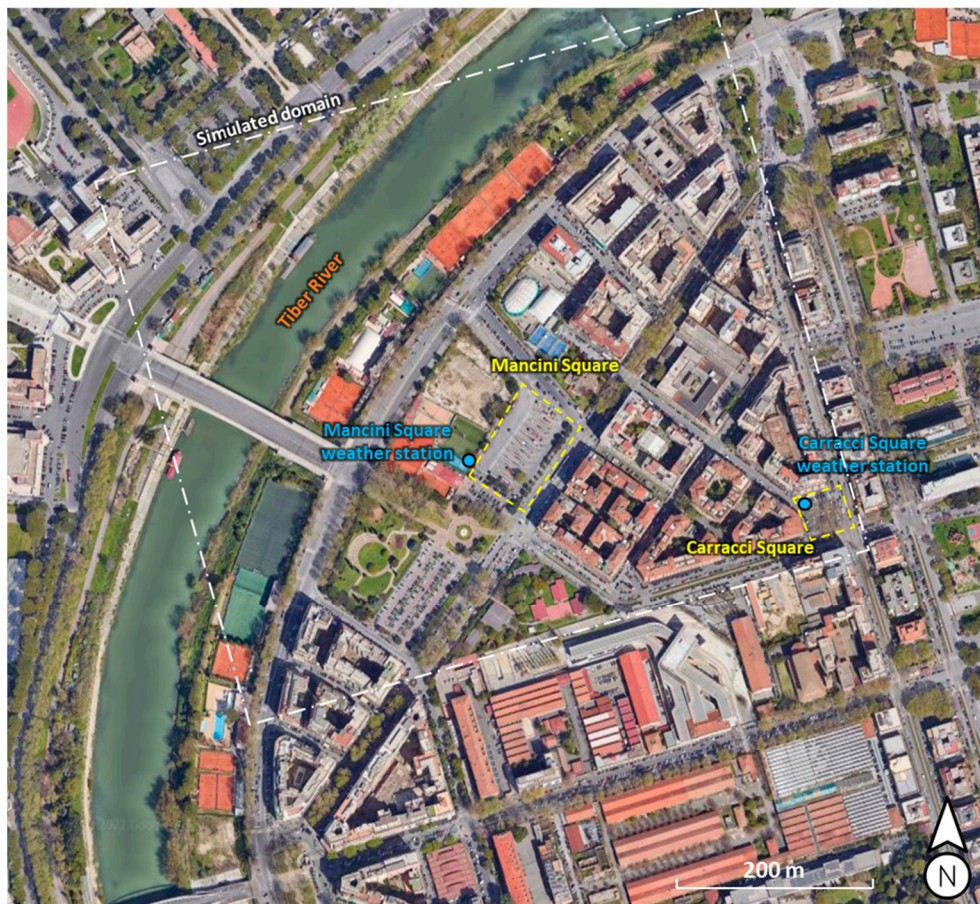

**Figure 2.** Aerial view for the simulated domain. Indication of Mancini and Carracci squares and weather stations positions.

*2.3. Mitigation Strategies*

The mitigation strategies were chosen following the best practices applied to other case study in literature, not considering the ones that cannot be applied due to urban restrictions. The ones selected for this research are:

1.    The use of cool pavements, with high values of albedo and permeability, with an albedo equal to 0.50;
2.    The use of grass pavers instead the traditional asphalt;
3.    Increase of vegetation, with species similar to those already present;
4.    Introduction of shading projecting roof in Mancini Square.

Pavements, in particular the asphalt, are one of the main factors for the increase of the urban heat island: in fact, there are often more present than green areas in cities. Asphalts with a high albedo led to a lighting save at night, given that the reflecting performance are higher than the ones with the asphalt usually used.

Cool pavements can be used to reduce the pavement solar absorption due to their better reflecting and permeability features than the common asphalt. Despite the common cool pavements adopted as usual mitigation strategy, there is the chance to use grass pavers as cool materials. These are porous pavements that are made of cool materials blocks that have a high solar reflectance, with an alternation of grass leading to an increase in terms of permeability and evapotranspiration effects.

About the shading projecting roof, it is worthy to observe that two advantages can be highlighted: (i) the shading effect to the ground that reduce the amount of solar radiation absorbed by the asphalt, and (ii) the possibility to install photovoltaic system for on-site electricity production.

Among the mentioned mitigation techniques, the "water mirrors" consist of the introduction of water sources inside the square in order to improve the cooling effects due to the evaporation of water. Furthermore, the total amount of solar radiation absorbed by the asphalt of Mancini Square is reduced due to the presence of these water mirrors.

### 2.4. Experimental Campain

In order to reach the most accurate model, two weather stations were installed in Carracci Square and in Mancini Square (see Figure 3). The numerical model was calibrated through the data logged by the weather station placed in Mancini Square.

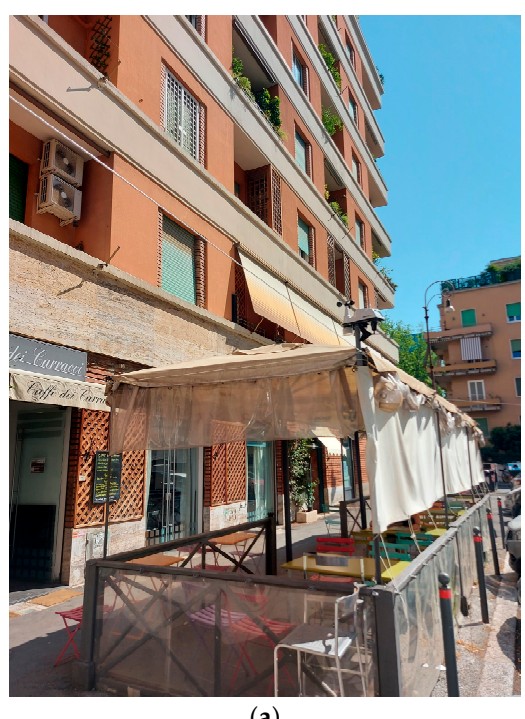 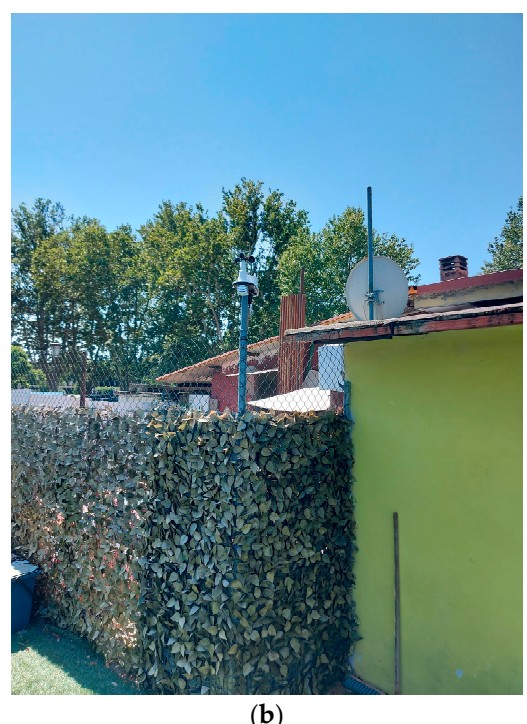

         (**a**)         (**b**)

**Figure 3.** Installation of the weather stations in Carracci Square (**a**) and Mancini Square (**b**).

The Davis Vantage Vue weather station is equipped with several weather sensors, including temperature, and humidity sensors, a pluviometer and an anemometer.

For the experimental analysis of pavements and walls surfaces, the portable Spechtrophotometer CM-2600d Konica Minolta was used. Measures were performed considering different combinations of colours, different values of roughness to analyse the different behaviour of the materials.

The weather stations were installed in 2021. Data were acquired from 21 July to 2 August, in order to collect data under different weather conditions. It was possible to observe that the weather station in Carracci Square is less exposed to the sun that the one in Mancini Square, and one day during which the data are similar will be chosen for the calibration of the numerical model.

### 2.5. Numerical Model Setup

The ENVI_met tool, which is based on the SVAT (Soil, Vegetation and Atmosphere Transfer) model, was used for the numerical model simulations. With the aid of this program, it is possible to model the microclimate of an urban region while taking into account all of the activities that take place in the soil, atmosphere, and integration of structures, surfaces, vegetation, water features, and pollution sources. The geometry characteristics and the corresponding numerical model are shown in Figure 4.

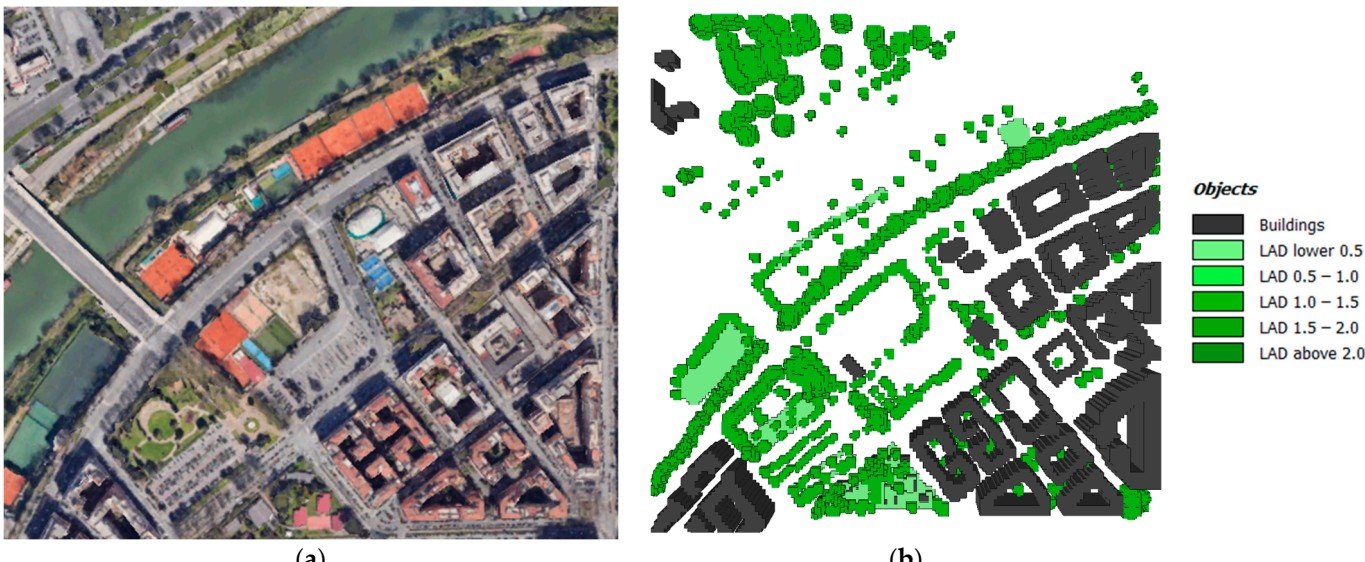

(**a**)                (**b**)

**Figure 4.** Comparison between the aerial view (**a**) and the ENVI_met model (**b**) of the simulated domain. ENVI_met. Green colors of (**b**) are referred to the vegetation Leaf Area Density (LAD).

The numerical simulations were performed for a total of 30 h starting from 6 pm of 21 July 2021 and ending of the midnight of 22 July 2021. The firsts 6 h of the simulations were used as a transitory period to have more accurately results for the whole 22 July day. Experimental data from Carracci square were used as the forced air temperature to the boundary of the numerical model due to its position on the numerical domain boundary (see Figure 2).

The domain is discretized with a quadrilateral grid imposed by ENVI_met. Due to the domain dimension (composed by a square area of 624 m on each side), it was necessary to balance the quality of the simulation and the time for the results generation. In particular, the grid consists of 156(x) × 156(y) × 30(z) cells with a grid size of 4 m and a time discretization of 10 s.

The developed model has buildings close to its boundaries, and the grid points cannot react on influences in the way grid points in the inner part of the model. For this reason, it is needed to add 5 nesting grids to the model. The Nesting Cells are usually used in every numerical model in which the simulation is not working reliably at their model borders and at the grids very close to them. These additional grids are used to move the numerical model borders as far as possible away from the area of interest in the core area. The reason for these problems is resulting from the fact, that the model cannot calculate real values for grid points along the borders.

## 3. Results and Discussion

### 3.1. Assesment of the Urban Heat Island Intensity

The assessment of the Urban Heat Island Intensity (UHII) is needed to justify the choice of the Mancini square as the object of the present study. As a matter of fact, to maximise the effect of mitigation techniques, the site chosen must have particular characteristics that bring to a very warmer area in which such as retrofit scenario can improve the air temperature level in the area.

It is possible to calculate the UHII as the air temperature difference of the examined area and a reference one that must be a rural area near the city. For the present research, were chosen the airport weather station of Ciampino as reference meteorological condition of a rural area near the city [58].

During the experimental campaign, the data collected has a similar trend and similar weather conditions for all the days. The period chosen for the experimental campaign represented the hottest days of the year. For this reason, it was taken one of these days as

a reference for the numerical simulation. In Figure 5 are shown the air temperature and relative humidity of the day 22 July that was taken in consideration for numerical model simulations. In Figure 5 were shown data from the two monitoring weather station of Mancini and Carracci square, while are also reported data acquired from the Ciampino airport that can be used as a reference of a rural area for the calculation of the Urban Heat Island Intensity (UHII).

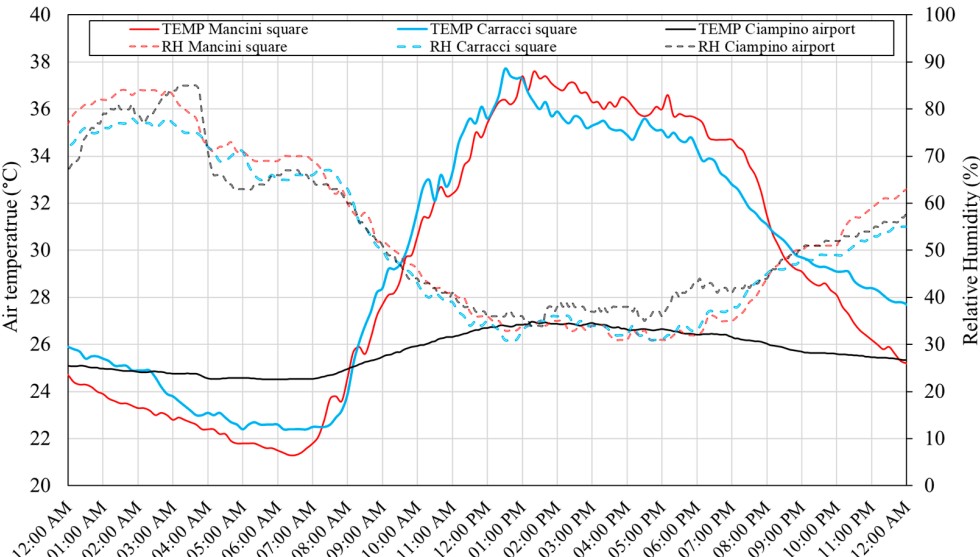

**Figure 5.** Air temperature and relative humidity trend during the measurement campaign on 22 July that was used for the numerical model simulations. Red line is referred to Mancini square weather station, Blue line is for Carracci square one and Black line is for data acquired from Ciampino airport.

From data of Figure 5 it is possible to notice that there is a warmer condition inside the Mancini square despite the data taken in Carracci one.

In Figure 6 is shown the maximum daily difference of the outdoor air temperature taken from Mancini and Carracci square and the Ciampino airport. Values of the Figure 6 are the daily UHII of Mancini and Carracci square and it is possible to notice that the days recorded with the experimental campaign represent the hottest part of the year. As a matter of fact, in the period of the experimental campaign the daily UHII vary from 2.1 to 5.8 °C for Mancini square, while are from 1.5 to 3.7 °C for Carracci square. It is possible to conclude that the Mancini square has the warmer condition compared to Carracci one that can justify the choice of this area as the object of this study.

For the present study, numerical simulation was done on the 22 July in which the data of Mancini and Carracci squares are similar.

### 3.2. Numerical Model Calibration

The first step of the numerical model implementation is reaching calibration of the results. In particular, the simulated and measured air temperatures and relative humidity were compared. The calibration step consists of checking, simulation after simulation, the relative error of simulated and measured data where the weather station of Mancini Square was installed. In each simulation, to minimize this error, it was corrected point by point the input data of the numerical model related to the air temperature and relative humidity. The difference between the experimental data and the simulated ones were assessed using two statistical indices: Mean Bias Error (MBE) and Mean Absolute Error (MAE). As results of the iterative calibration method described above, it was calculated the MBE and MAE errors for each simulation. The calibrated model reached a MAE value of 0.29 and a MBE equal to 0.06.

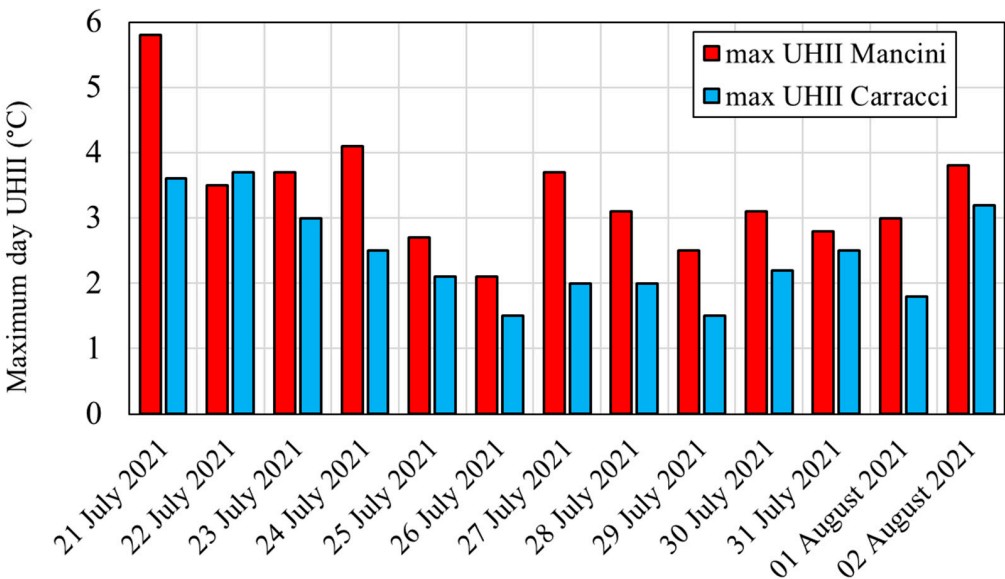

**Figure 6.** Daily UHII of Mancini and Carracci square taken data from Ciampino airport as a reference.

### 3.3. Numerical Models of Mitigation Strategies

In evaluating the effectiveness of urban heat island mitigation interventions, numerical simulations were carried out by combining compatible mitigation techniques to obtain a heat island mitigation scenario as effective as possible.

Different type of mitigation strategies inside the Mancini Square were implemented (see Figure 2) and configurations are shown in Figure 7.

The analysed original scenario without any mitigation techniques is named Scenario 1. In this case wide asphalt zones are employed as sidewalks and pavement. Additionally, there are low trees that do not provide enough shade inside Mancini Square. Furthermore, in the proximity of the square there is an high traffic road and two parking area characterized by few trees.

Scenario 2 is related with the introduction of cool materials inside the square. In particular, it was used a cool pavement with an albedo equal to 0.5. As shown in Figure 7, the intervention area is highlighted with the dotted red line and correspond to the whole square.

The adoption of a grass pavers as the pavement of the square is related to the Scenario 3. This case has the same configuration of the standard Scenario 1 with only the use of different type of pavements.

The improvement of the vegetation is considered in the Scenario 4. In particular, medium and tall trees and grass have been added in the centre of the square. Furthermore, it was improved the amount of trees in the park adjacent to the square as shown inside the red dotted line of Figure 7.

Finally, in Scenario 5 it was used a shading projecting roof as a mitigation techniques as shown in Figure 7. This case has the same configuration of the standard Scenario 1 with only the introduction of a shading projecting roof in the centre of the square.

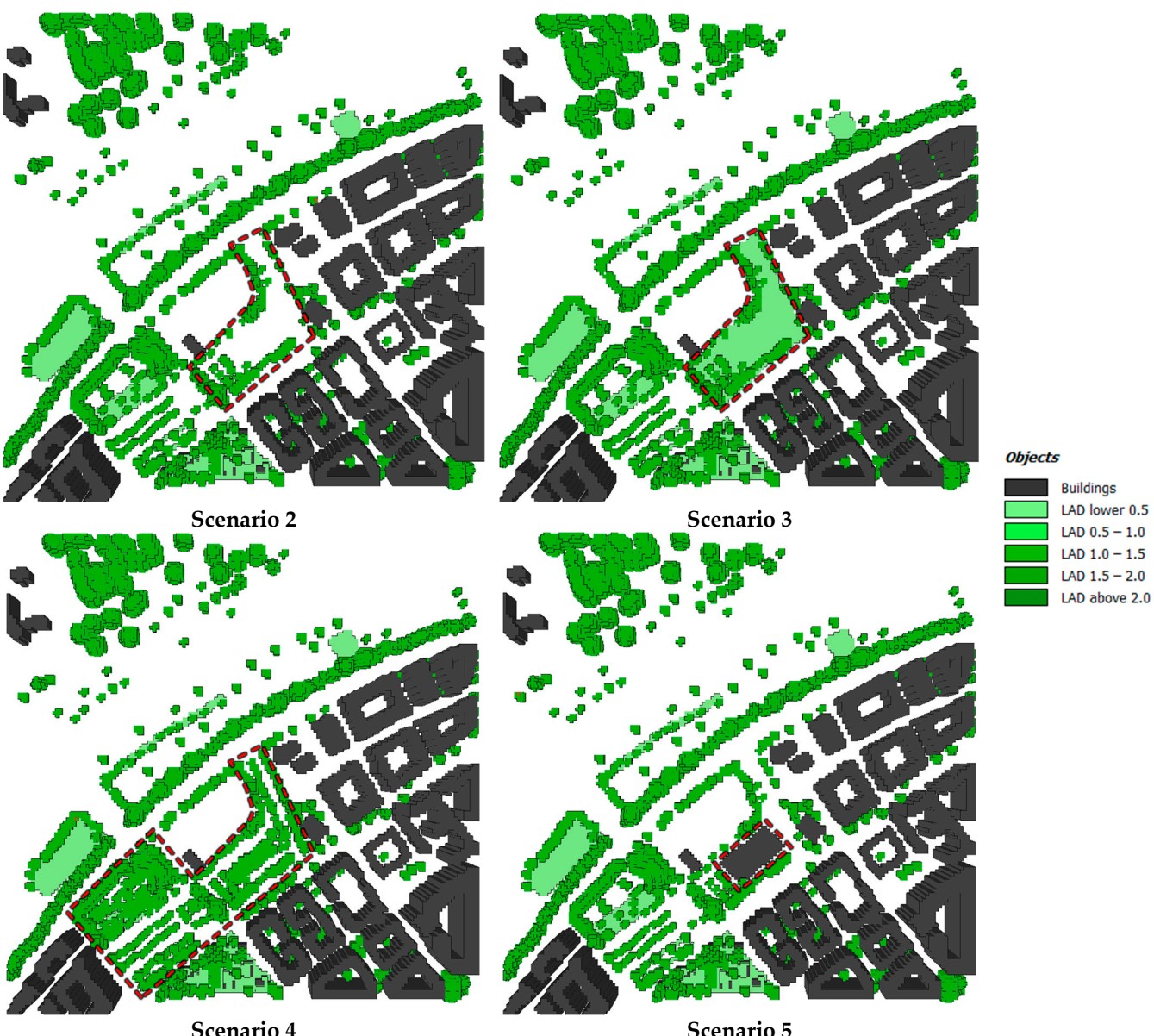

**Figure 7.** Numerical model used for the mitigation scenarios. Green colors are referred to the vegetation Leaf Area Density (LAD). The interested area is inside the dotted red line.

### 3.4. Air Temperature Spatial Variation

The results related to the hottest hours of the day are presented in Figure 8, at 1 pm and at an altitude of 1.75 m from the ground, comparable to the average height of people. Figure 8 presented the air temperature difference taking as a reference the scenario 1 that represents the actual situation of the area and are shown the related results to the scenarios 2, 3, 4 and 5 of the mitigation strategy techniques applied.

Considering the results shown in Figure 8 related to the Mancini Square, a maximum decrease in the air temperature of 1.2 °C was obtained through the application of cool material. This reduction reaches up to a maximum of 2.88 °C when a grass paver is simulated. This is due to the advantages of combining cool material of the pavement block and the evapotranspiration effect of the grass.

The improve of the vegetation of Scenario 4 bring to a maximum decrease of the air temperature of 1.46 °C. This value is less than the installation of grass pavers of Scenario 3 (in which there is the use of vegetation like grass), because the area covered by the vegetation of Scenario 4 is confined in the centre of the square while in the Scenario 3 the

area interested by the mitigation strategy is all the square. As a matter of fact, analysing the data shown in Figure 8 is possible to notice that the air temperature decrease is in confined areas despite of the data taken from Scenario 2, 3 or 5.

The use of the shading projecting roof in the square allows an air temperature reduction up to a maximum of 2.07 °C but, it is worthy to notice that the positive effect is strictly related to the area under the shading roof. Outside this area, the air temperature is higher due to the low albedo of the original asphalt.

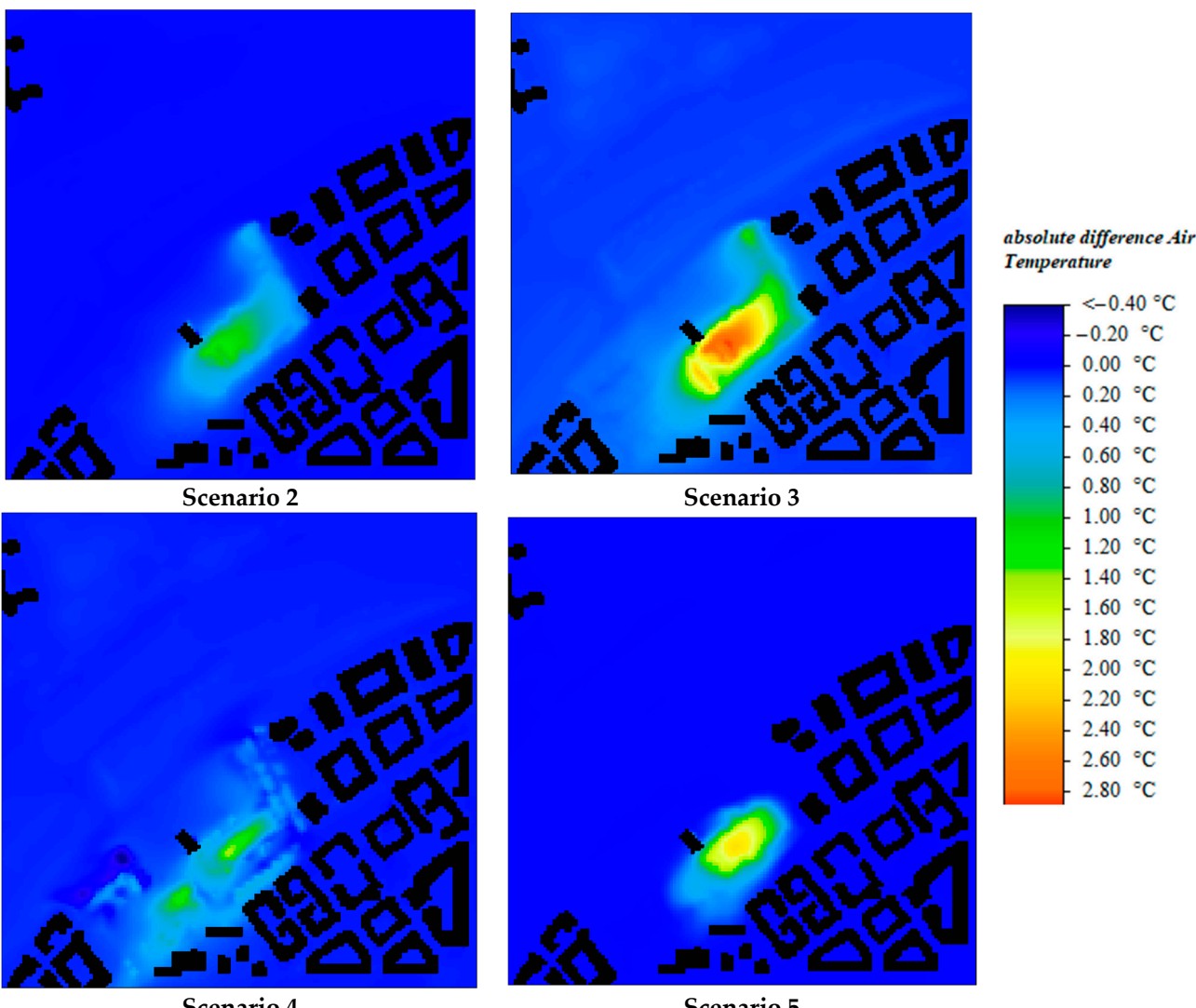

**Figure 8.** Comparison of Scenario 1 and Scenarios 2, 3, 4 and 5: simulation of air temperature at 1 pm and at an altitude of 1.75 m.

### 3.5. Time Variation of Air Temperature

A comparison of the estimated data was done in the three distinct places in the simulation domain depicted in Figure 9a to examine the impact of the various mitigation approaches. This quantitative analysis shows result during all the hour of the day highlighting the less or more effects of the mitigation strategy during the day.

In Figure 9b–d are shown the air temperature differences in different hours of the day, calculated in three points (named receptors R1, R2 and R3) at a height of 1.75 m above the ground. The receptor R1 is very close to the Mancini Square; the receptor R2 is related to a point in the middle of the square; while R3 represent a place far from the requalified area of Mancini Square and placed in the street that boards the square.

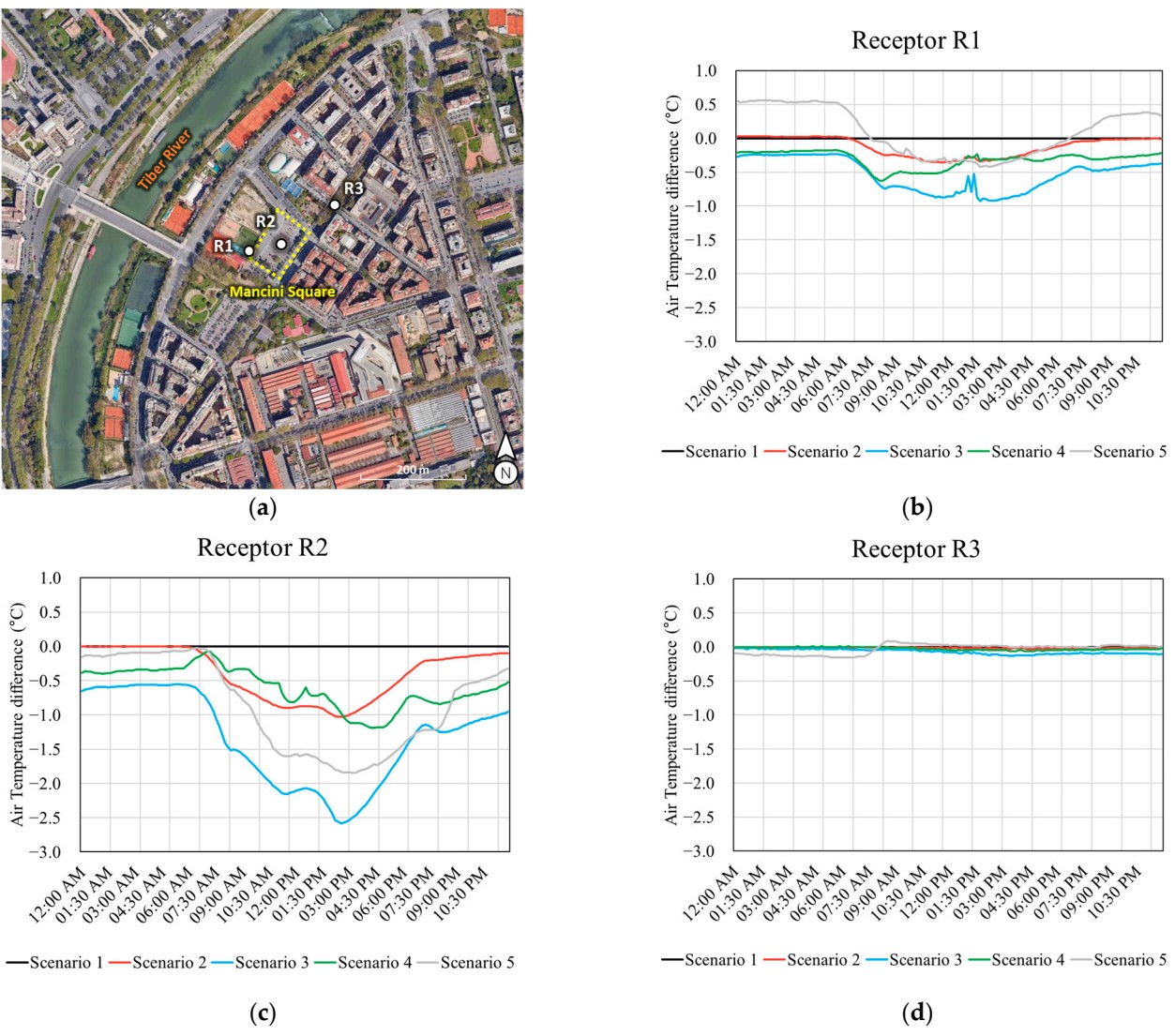

**Figure 9.** Comparison of the Scenarios 2, 3, 4, 5 and the actual situation (Scenario 1) in terms of air temperature difference at an altitude of 1.75 m considering Scenario 1 as reference. The points of evaluation are shown in (**a**) while results are shown in (**b**–**d**).

In Figure 9d are shown the air temperature difference on the receptor R3 of the different mitigation scenarios and the actual situation (scenario 1). It is possible to notice that the effects of the different mitigation techniques are negligible far from the square. As a matter of fact, the whole trend has a temperature difference close to zero compared to the original scenario (Scenario 1).

In Figure 9b,c represent the results near or inside the retrofitted area respectively. It is worth to notice that there are more effects of the mitigation techniques when the data are analyzed inside the area redeveloped, that is in the receptor R2 that is placed in the center of the Mancini square. Instead, less effects can be found in areas near the examined site, that is in the point R1 located in the proximity of the Mancini square.

Considering the introduction of a shading projecting roof in the centre of the square, is possible to confirm the results previously discussed in Section 3.4. As a matter of fact, comparing the results in points R1 and R2 (see Figure 9b,c), the trend is negative all hours a day only in the in R2 where the point is placed in the same place of the installation of the shading projecting roof.

Regarding the use of cool materials and grass pavers, it was confirmed the decrease of air temperature with more advantages with the grass pavers mitigation scenario. Fur-

thermore, there are more positive effects in the R2 place than in the R1 one, due to the high presence of asphalts materials in the middle of the square compared to the area taken into consideration for receptor R1. Finally, the adoption of an extensive mitigation strategy has the higher effects inside the area interested of the redevelopment. In the proximity of the objected area, the air temperature has a slightly reduction compared to the ones obtained inside the area of interest.

About the improvement of vegetation with medium and tall trees and grass placed in the middle of the square, it is possible to notice more relevant air temperature decrease near the improved vegetation area. However, the adoption of vegetation inside the square can reduce the air temperature near this one as shown in the results of receptor R1. As a matter of fact, also in R1 the air temperature difference has a negative value all hours of the day due to the replacement of an asphalt area of the square with vegetated one.

Finally, considering the results in the middle of the square (data from receptor R2 of Figure 9c) the more effective techniques considering the average difference in the entire simulated day are the use of grass pavers of Scenario 3 ($-1.35$ °C), followed by the use of shading projecting roof of Scenario 5 ($-0.85$ °C), the improve of vegetation of Scenario 4 ($-0.60$ °C) and the cool material installation of Scenario 2 ($-0.39$ °C). Furthermore, considering the maximum air temperature difference with the actual situation (Scenario 1) it was recorded a maximum reduction of 2.58 °C at 2:30 pm with the adoption of grass pavers, of 1.85 °C at 3:10 pm with the adoption of shading projecting roof, of 1.19 °C at 4:10 pm with the use of vegetation and 1.03 °C at 2:30 pm with the installation of cool material.

## 4. Conclusions

In this work different types of technical options for reducing the microclimatic conditions in a heavily populated Roman neighbourhood were investigated. In order to evaluate the impact of the mitigation strategies in terms of an air temperature reduction, a numerical model was developed through the ENVI_met tool. To establish the boundary conditions and calibrate the numerical model, experimental data of the air temperature and relative humidity were collected.

A methodology was developed to assess the urban heat island effect in a localized urban area and to evaluate the effects of different kind of mitigation strategies. The numerical model was calibrated with weather data measured inside the examined area and near the simulated domain. Furthermore, it was used different measurement sensors for accurate calibration of the model, with a measure of air temperature, relative humidity, solar radiation, rain, wind velocity and direction, and surface albedo. This procedure overtaking the limit of the software in reproducing accurate weather conditions without calibration. Finally, combination of extensive mitigation strategies (cool pavements, greenery, grass pavers) with local strategies (shading) were investigated.

The methodology is useful to analyse air temperature conditions in warmer urban areas. For this reason, the assessment of the Urban Heat Island Intensity (UHII) is needed to justify the choice of a particular site as the object of the scientific study on UHI. As a matter of fact, to maximise the effect of mitigation techniques, the site chosen must have particular characteristics that bring to a very warmer area in which such as retrofit scenario can improve the air temperature level in the area.

It is possible to calculate the UHII as the air temperature difference of the examined area and a reference one that must be a rural area near the city. For the present research, were chosen the airport weather station of Ciampino as reference meteorological condition of a rural area near the city. It was possible to conclude that the Mancini square has the warmer condition compared to Carracci one that can justify the choice of this area as the object of this study.

After analysing the results of the interventions, it is concluded that an intervention of the pavement albedo implies an improvement of the conditions associated to thermal stress and, in particular, the application of the grass pavers would lead to the greatest

benefits. Furthermore, the improvement of vegetation inside the Mancini Square can lead to a reduction of the air temperature during all hours of the day both for areas placed inside and very close the square.

As far as the shading canopy is concerned, there is an important local effect on the climatic conditions, an effect which however decrease as people move away from the shadows. When evaluating the shading shelter, it is important to take in consideration that there is a further advantage: in addition to the shading effect, there is the possibility to instal photovoltaic panels to produce renewable energy.

Therefore, the chosen heat island mitigation techniques make it possible to decrease the air temperatures at the street level, with interventions able to reduce the air temperature up to 2.58 °C at the centre of the square, and up to 0.93 °C in placed very close to the square, during the hottest hour of the day.

**Author Contributions:** Conceptualization, G.B. and E.d.L.V.; methodology, G.B. and E.d.L.V.; software, G.B.; validation, G.B.; investigation, G.B. and E.d.L.V.; resources, G.B., E.d.L.V. and R.d.L.V.; writing—original draft preparation, G.B., E.d.L.V., L.E. and R.d.L.V.; writing—review and editing, G.B., E.d.L.V., L.E. and R.d.L.V.; supervision, R.d.L.V. and L.E. All authors have read and agreed to the published version of the manuscript.

**Funding:** This research received no external funding.

**Institutional Review Board Statement:** Not applicable.

**Informed Consent Statement:** Not applicable.

**Data Availability Statement:** Not applicable.

**Acknowledgments:** The authors would like to thank the contribution of the Department of Architecture of the University of Roma Tre team headed by Paolo Desideri in which there was the participation of Francesca Romana Cattaneo, Roberto D'Autilia, Giorgia De Pasquale, Alessandro Gabbianelli, Luca Montuori, Enrico Nigris, Maria Pone and Matteo Staltari. The Department of Architecture of the University of Roma Tre team has done a collaboration to the project development with the Department of Industrial, Electronic and Mechanical Engineering research team headed by Roberto De Lieto Vollaro in which there was the participation of Gabriele Battista, Emanuele De Lieto Vollaro and Marco Formiconi.

**Conflicts of Interest:** The authors declare no conflict of interest.

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
