# Peer review of "Urban Overheating Mitigation Strategies Opportunities: A Case Study of a Square in Rome (Italy)"

_sustainability, doi:10.3390/su142416939_

Round 1

Reviewer 1 Report

Review report

Manuscript: Urban overheating mitigation strategies opportunities: a case study of a square in Rome (Italy)

Journal: Sustainability, MDPI

---

The manuscript, as in detail above, adopted ENVI-met, a simulation model for urban microclimate, to evaluate four alternative solutions for urban heat island (UHI) mitigation in Mancini Square, Italy. The content of this manuscript seems interesting to readers, and it potentially contributes to urban heat mitigation strategies. Yet, it needs major revision and clarification the entire manuscript before it could be published in the Journal.

Some of the major comments are given below:

1. The Introduction should be clarified and better reorganized. There is too much information from various pieces of literature, but it is less associated with each other. It should keep the main flow directly linked to the main points of research objectives.

Authors missed crucial controlling factors of UHI related to land configurations assessed by landscape metrics as mentioned in many published works, such as:

·        Influences of buildings on urban heat island based on 3D landscape metrics: an investigation of China’s 30 megacities at micro grid-cell scale and macro city scale

·        How do disparate urbanization and climate change imprint on urban thermal variations? A comparison between two dynamic cities in Southeast Asia.

·        Effects of landscape composition and pattern on land surface temperature: An urban heat island study in the megacities of Southeast Asia

Also, mitigation solutions need to be more emphasized and in a systematic maner because this is the key point of this research. Authors may find useful information via some related reviews:

·        Recent development and research priorities on cool and super cool materials to mitigate urban heat island

·        Using cool pavements as a mitigation strategy to fight urban heat island—A review of the actual developments

·        A review on the development of cool pavements to mitigate urban heat island effect

·        Cool pavements for urban heat island mitigation: A synthetic review

Lines 61-63, it should be reconsidered as the high population is related to anthropogenic heat sources contributing to UHI.

2.  Methodology requires major clarification since it is unclear in its present form, such as, what are the inputs and outputs of the model? How do input data?

It should be better organized. For example, most of section 3.1 should be placed in methodology.

Besides, the scenarios have to be described in detail, such as the planned area, and location. Authors may give a related sketch, spatial planning, etc. The reviewer was concerned that the differences in a temperature reduction of different scenarios might come from disparate locations and usable area of each scenario instead of their nature.

3. Results and Discussions:

·        Section 3.2: Actually, the spatial difference between scenarios tends to be more localized. Therefore, temperature difference maps may better display the scenario performance rather than the current Fig. 5.

·        Section 3.3: Authors should be noted that analyzing temperature at the specific points only reflects temperature fluctuation at this specific location rather than the effect of the proposed solution. Plus, ambient temperature is more meaningful to human-well being, what we wish. Therefore, authors are recommended to assess temperature changes in different zones under scenarios that may limit the extremity at a specific point.

·        The manuscript also requires more discussions drawn from the research findings.

4. Besides, the manuscript has to check its writing carefully as there are many typos and unclear contents. 

Reviewer 2 Report

General comment: The study is generally well presented and it leads to possible solutions for UHI mitigation in a limited zone in an urban area. However, there are some aspects that would need clarifications, like state of the art knowledge on UHI characteristics in European urban areas, characteristics of UHI in the selected area, justification for numerical simulations carried for only one day. The manuscript would also benefit from improving the use of English language, as at present it may be misleading and difficult to follow in some parts.

Detailed comments:

- most of the cited literature in relation to UHI (lines 50-78) refers to non-European cities, which may not be fully relevant for the case discussed. Please complete this section with an overview of UHI characteristics in Europe.

- the second meteorological station (in Carracci Square) is mentioned, but apparently its measurements were not used in the study. Please detail/explain the relevance of this station in the context of the study or, if it’s not the case (e.g., its measurements were not used) please remove it from the text.

- please provide an overview of UHI characteristics at the selected location, to better justify its selection (e.g., based on measurements from the weather station placed in Mancini Square).

-please justify the selection of a particular day for the simulations. Also, please justify why the simulations for only one day are enough to be representative and conclusive for the effect of different UHI mitigation measures.

- Examples on the need to rephrase/improve the use of English:

lines 31-32: ‘

·        ‘People moved from the countryside to the cities all over the world, but not in the same way ‘ –maybe not at the same rate?

·       ‘In the last few centuries, the growth has started to go faster

lines 35-36: ‘Urban Heat Island  stands for a microclimate phenomenon of the cities’ – meaning that UHI describes/relates to ? if so please rephrase, otherwise please clarify.

lines 110-111: ‘A computer model of a university campus in Rome was created by Salata et al. [51] and may be used to examine various UHI mitigation tactics.’ – the cited article says that they used/employed ENVI software, thus they did not ‘create’ a model, but they investigated certain aspects with the use of numerical simulations done with ENVI software. This is totally different from ‘creating a computer model’ which generally means to develop the software itself. Furthermore, those simulations were used in their analysis/article, thus they cannot be used for other UHI mitigation tactics or on another site/location, as for these new simulations would be necessary. Please reformulate the paragraph to be clear and accurate.

lines 199-201: ‘I must  be observed that the weather station in Carracci Square is less exposed to the sun that the one in Mancini Square, and one day during which the data are similar will be chosen’  - probably a typo ‘I’ instead of ‘It’; the day will be chosen or has been chosen? In any case, please describe the selected day from meteorological points if view, including data from the weather station(s) used.

line 204: ‘2.5. Nemerical model setup’ – probably a typo-  ‘Nemerical’ instead of ‘Numerical’

lines 205-206: ‘The ENVI_met tool, which is based on the SVAT model, was used to create the numerical model (Soil, Vegetation and Atmosphere Transfer).’ – From my understanding based on the rest of the manuscript, the ENVI software was used for numerical simulations on the selected area, which is different from creating a ‘numerical model’(meaning to develop the software itself). Furthermore, ‘(Soil, Vegetation and Atmosphere Transfer)’ is the full name of SVAT model and not the name/meaning of the ‘numerical model’ created by the authors; thus, this parenthesis should be located immediately after the acronym SVAT.

line 221: ‘it is needed to add 5 nesting grids to the model’ – from my understanding, the 5 added grids (in fact grid points/cells) are  the ‘buffer’area (i.e., a zone around the area of interest, which is not used in the analysis, but it is necessary in order to allow the ‘adaptation’ of parameters from the larger-scale input to the finer-scale area of interest); the buffer area is present in almost any limited area numerical simulation; ‘nesting’ has a totally different meaning e.g. in numerical weather prediction models, see for example (Gómez-Navarro et al, 2015). Please rephrase such that to be clear and accurate.

line 262: ‘at an altitude of 1.75 m from the ground, comparable to the average altitude of people’

References: Gómez-Navarro, J. J., Raible, C. C., and Dierer, S.: Sensitivity of the WRF model to PBL parametrisations and nesting techniques: evaluation of wind storms over complex terrain, Geosci. Model Dev., 8, 3349–3363, https://doi.org/10.5194/gmd-8-3349-2015, 2015.

Reviewer 3 Report

Q1. The detailed content and contributions of the proposed method have not appeared in the Abstract so that it is hard to know the contributions of the paper through the Abstract?

Q2. Literature review should be improved highlighting the novelties and the main contribution.

Q3. It is recommended to put some latest research on Impacts of UHI on Rome’s climatic condition in the literature section.

Q4. Simulation results are unclear and should be discussed clearly.

Q5. The conclusion section is expected to show how the work advances the field from the present state of knowledge. The author should provide a clear scientific justification for this work in conclusion section and indicate uses and extensions if appropriate.

Q6. Check the grammatical and typo errors and correct them.

Round 2

Reviewer 1 Report

The reviewer highly appreciated the authors’ efforts to revise the manuscript with the comments from all reviewers. The quality of the manuscript has significantly improved compared to the previous version. However, It needs to be reconsidered on some of the following issues:

1.     Abstract should be better improved with some specific numbers to highlight the effects of the proposed strategy.

2.     In the first round, the reviewer debated that the authors did not mention about influences of landscape configurations on urban heat islands. It is closely associated with this study, so the authors should introduce it in the Introduction and the Scenarios also.

For example, in Scenario 4 (improvement of vegetation), the cooling effect would be different, if we plant the same quantity of trees with different arrangements.

Moreover, detail descriptions of scenarios are required, e.g., area, arrangement, ….

Figure 7 should highlight the specific planned zones.

3.     The result of urban heat island intensity presented in Section 3.1, however, the corresponding method to analyze it could not find in the Methodology section.

4.     Please add legend for Figures 4-b and 7.

Reviewer 2 Report

Thanks to the authors for their work. The manuscript has been significantly improved becoming more clear and better structured,  thus exploiting and emphasizing the potential of the study. Only minor typos should be revised.  
